# Infant Feeding Choices during the First Post-Natal Months and Anthropometry at Age Seven Years: Follow-Up of a Randomized Clinical Trial

**DOI:** 10.3390/nu14193900

**Published:** 2022-09-21

**Authors:** Hans Demmelmair, Manja Fleddermann, Berthold Koletzko

**Affiliations:** 1Dr. Von Hauner Children’s Hospital, University of Munich Medical Centre, Ludwig-Maximilians-Universiät München, 80337 Munich, Germany; 2HiPP GmbH & Co. Vertrieb KG, 85276 Pfaffenhofen, Germany

**Keywords:** infant formula, follow-up, α-lactalbumin, height, weight, body mass index

## Abstract

The Belgrade–Munich Infant Milk Trial (BeMIM) randomized healthy term infants into either a protein-reduced intervention infant formula (IF) group, with an α-lactalbumin-enriched whey and long-chain polyunsaturated fatty acids, or a control infant formula (CF) group. A non-randomized breastfed group (BF) was studied for reference. We assessed the long-term effects of these infant feeding choices on growth measures until the age of seven years. Weight, standing height, head circumference, and percent body fat (using skinfolds and bioelectrical impedance) were determined with standardized methods. A total of 161 children out of the 256 completers of the initial study (63%) participated in the seven-year follow-up. Children in the three study groups did not differ in their anthropometric measures, including body mass index (IF 16.1 ± 2.6, CF: 15.6 ± 1.7, BF: 15.6 ± 2.5 kg/m^2^, mean ± SD). IGF-1 serum concentrations determined at the age of 4 months contributed to explaining the variances in weight (*p* = 0.001), height (*p* = 0.001) and BMI (*p* = 0.035) z-scores at the age of seven years, whereas insulin levels at four months did not. Different feeding choices during the first four months of life leading to higher energy efficiency and increased growth with IF did not affect later growth outcomes at an early school age. Diet-induced modulation of IGF-1 in the first months of life may have lasting programming effects on later growth.

## 1. Introduction

Pre- and post-natal diet influences the long-term development of infants, including anthropometric development and the long-term obesity risk [1]. Observational studies have consistently associated rapid early weight gain with an increased risk of later obesity [2]. Several cohort studies established that infant formula feeding, compared with breastfeeding, is associated with a higher risk of rapid early weight gain, as well as later obesity [3,4,5]. The protein content of infant formulas is generally higher than the protein content of human milk, and both randomized clinical trials and cohort studies have associated a high infant protein intake with an increased risk of later childhood obesity [6]. The positive association of weight gain with protein intake in infancy was shown in the CHOP study, where 1090 infants in five European countries were randomized into higher or lower protein formula groups for the first year of life. Children assigned to the higher protein group had significantly higher BMI z-scores at the later ages of 2 years (0.40 ± 0.95 vs. 0.19 ± 0.89, M ± SD) and 6 years (0.55 ± 1.29 vs. 0.25 ± 1.12) [7,8]. The anthropometric development of the lower protein group was similar to the breastfed reference group. At the age of six months, serum indispensable amino acid concentrations in the lower protein group were more similar to those of the breastfed infants but lower than in the infants in the higher protein group [9]. However, regression analyses did not show significant associations of any of the measured amino acids at the age of six months with the obesity risk of these infants at the age of six years [10]. Nineteen other metabolites measured at the age of six months were associated with weight gain from birth to age six months, but only 1-myristoyl-2-hydroxy-phosphatidylcholine levels predicted obesity risk at the age of six years, which suggested a complex mechanism relating early diet to later anthropometric development [10].

The monocentric Belgrade–Munich Infant Milk Trial (BeMIM) has a similar design as the CHOP study, with double-blind randomization of formula-fed infants into higher- or lower-protein-formula groups and an observational breastfed group. However, the formula intervention lasted only until four months of age. This study’s primary objective was to assess the growth until the age of four months [11]. The intervention formula (IF) had a lower protein and carbohydrate, but higher fat content, than the control formula (CF) [11]. The addition of α-lactalbumin and long-chain polyunsaturated fatty acids to IF and different free amino acids supplementation of the two formulas led to differences in the amino acid and fatty acid composition, as previously reported [11]. In the children that completed the intervention study, the daily weight gain from postnatal day 30 to 120 was not significantly different between the formula groups, but the daily length gain was significantly higher in the lower protein IF group (0.11 ± 0.02 vs. 0.10 ± 0.02 cm/day) [11]. Serum amino acid concentrations at four months of age differed between the groups, but associations with insulin and IGF-1 were limited and tended to be stronger with insulin for most amino acids [12]. A follow-up examination at four years of age did not show any significant group differences in height, weight and body composition measures [13]. Although some of the metabolites measured in infants at the age of four months were related to growth during the intervention period, none of these metabolites predicted the anthropometry at the age of four years [13].

No definite mechanism mediating the association between early protein intake and later anthropometric development, including obesity risk, has been identified. The “Early protein hypothesis” posits that high infant protein intakes lead to increased circulating levels of branched-chain amino acids, IGF-1 and insulin, which could be associated with anabolic effects via mTOR or other mediators [5,14]. Another suggested mechanistic route is that higher growth in the early postnatal period is associated with higher weight and BMI in later periods of childhood and adult life [2,15].

We aimed to evaluate the effects of infant feeding on later anthropometry with potential mediating effects of early insulin and IGF-1 secretion in the BeMIM study. Anthropometric data were collected with an infant feeding intervention with different infant formulas or breastfeeding at the ages of one and four months, with a 4-year follow-up [11,13] and newly collected measures at the age of seven years. This enabled the evaluation of long-term effects of feeding and hormone levels at the age of four months on later anthropometry.

## 2. Materials and Methods

The study population of the 7-year BeMIM follow-up consisted of children that completed the BeMIM study at the age of four months, as previously reported [11]. Infants were included if they had completed the main study, had participated in the examinations at the ages of one and four months, and participated in the 7-year follow-up (Table 1). The main analysis of the data was performed with the intention to treat the population, which comprised all subjects who were enrolled, including those with minor and severe non-compliance to protocol and subjects with missing values, if anthropometry at the ages of one and four months was available. Comparisons of the anthropometric development were repeated for the per-protocol population, which included all subjects without severe noncompliance with the protocol.

The study protocol was reviewed by the Ethical Committee of the Clinical Centre of Serbia in Belgrade/Serbia (Broj 2494/26). Signed parental informed consent was obtained for all infants. The trial was registered at Clinical trials.gov (NCT01094080).

### 2.1. Anthropometric Measurements

Anthropometry, including weight, height, head circumference, and body fat content measured via skinfolds and bioelectrical impedance, was undertaken at seven years of age by trained pediatricians. Weight was determined with a Seca 711 scale (Seca, Hamburg, Germany) equipped with a measuring rod (Seca 220, Hamburg, Germany) to measure height. Head circumference was measured with a tape (Seca 212, Hamburg, Germany). The weight and head circumference measurements were performed in duplicate and documented with an accuracy of 10 g and 0.1 cm, respectively. The equipment was checked and calibrated every two months to ensure the accuracy of the measurements. The z-scores for age were calculated relative to the growth standards of the WHO for children (https://www.who.int/tools/growth-reference-data-for-5to19-years, accessed on 18 Setember 2022) using the WHO Anthro-Plus SAS macro (https://www.who.int/tools/growth-reference-data-for-5to19-years/application-tools, accessed on 18 September 2022).

Skinfolds were measured using a Holtain caliper (Holtain Ltd., Crymych, UK) at the left body axis at two sites (triceps and subscapular). All measurements were performed in triplicate and documented with an accuracy of 0.2 mm. Body fat percentage was calculated via predictive skinfold equations [16]. As a further validated method for the determination of body composition, bioelectrical impedance analysis (BIA) was performed in duplicate using a BF 906 Body Fat Analyzer (Maltron, Rayleigh, United Kingdom) [17].

The 7-year follow-up included questions and data retrieval from medical records to determine the presence of serious illnesses in the children that could have influenced growth. These illnesses were considered serious adverse events and were used to assess the long-term safety of the study formulas.

### 2.2. Statistics

Statistical analyses included data collected in the main study, in the 4-year follow-up, the insulin and IGF-1 measured at the age of four months [11,12,13], and the newly collected data at the age of seven years. Continuous data are presented as the mean ± SD or median and interquartile range formats (IQR, 25th and 75th percentile). Categorical variables are presented as the count and percentage format based on valid cases.

Besides the absolute anthropometric measurements, daily gain parameters of anthropometric measurements of weight (g/d), length (mm/d) and head circumference (mm/d) were derived for the observation periods of one to four months, one month to seven years, four months to seven years and four to seven years. Changes in z-scores between four months and one month, seven years and one month, seven years and four months, and seven and four years were evaluated comparable with the absolute anthropometric parameters.

Pearson’s chi-square test was used for the comparison of categorical data between groups. Student’s *t*-test (Satterthwaite for unequal variances) was used for the comparison of normally distributed continuous data. Correlations were evaluated using Fisher’s adjusted Spearman correlation.

ANCOVA was applied to estimate the effect of formula type on the absolute or z-score anthropometric outcome while including the child’s age at the seven-year visit, sex, age of the mother at the seven-year visit and the mother’s smoking status at the seven-year visit as confounders. The corresponding baseline value (month one) of the anthropometric parameter under investigation was included if available. Adjusted differences are presented as least-squares means and corresponding 95% confidence intervals between formula groups. Separate linear regression models, including corresponding 1-month z-scores, were applied to test for an effect of IGF-1 and insulin at four months of age on the z-score anthropometry at the age of seven years.

Significance was accepted at a *p*-value < 0.05 without considering adjustment because of multiple testing. The results as presented should be seen under an explorative perspective as the study was powered based on the main outcome [11]. Statistical analyses were performed with World Programming System WPS Software, Ver. 4.1 (London, UK) using SAS language and SPSS Ver. 26 (IBM Corp., Armonk, NY, USA).

## 3. Results

Two-hundred and fifty-six children, who were included in the data analysis of the main study, were invited to participate in the 7-year follow-up study and 161 children participated (63% participation; 54 IF (63.5%), 50 CF (60.9%) and 57 BF (61.9%) (Figure 1).

No differences in the demographic characteristics were found between the formula groups at the age of seven years, while BF infants differed from formula-fed infants in the maternal educational status, as was previously observed at baseline (Table 1). The overall mean age of the children at the 7-year follow-up was 6.99 ± 0.04 years and did not differ between the groups.

A comparison of the families who participated in the 7-year follow-up with the non-participants revealed that in most aspects, both groups were not significantly different. Significant differences were only seen for maternal education and age, and the percentage of boys tended to be higher among the participants (Appendix A).

The anthropometric outcomes at the age of seven years are reported as absolute values and z-scores in Table 2. Values for height, weight, head circumference, BMI, skin folds and body fat percentage were not significantly different between the groups according to t-tests and ANCOVA. Findings in the per-protocol population were very similar to the intention-to-treat population (Appendix A). Body fat percentages determined via bioelectrical impedance and estimated from skinfolds were highly correlated (r = 0.62, *p* < 0.001). With both methods, significantly higher percentages for girls were observed (bioelectrical impedance: 16.2 ± 8.5 vs. 12.9 ± 6.1%, *p* = 0.005; skinfolds: 19.1 ± 5.5 vs. 16.3 ± 4.9%, *p* < 0.001). The weight, length and BMI developments of the children participating in the 7-year follow-up from the start of the main study (month 1) until the 7-year follow-up are shown as z-scores in Figure 2, Figure 3 and Figure 4.

Growth during the different periods is shown as changes in absolute values for weight, length/height and head circumference (Table 3). There were significant differences between the formula groups and between IF and BF during the first four months of life for weight and length gain for the follow-up participants. There were no significant group differences after the age of four months, except for a greater head circumference gain from four years to seven years in CF than BF. Corresponding developments in the weight, length and BMI z-scores are shown in Appendix A.

Cole et al. established child overweight and obesity criteria, which relate a BMI of 17.9 kg/m^2^ in seven-year-old boys (17.8 in girls) to the adult overweight cut-off BMI of 25 kg/m^2^ and 20.6 kg/m^2^ for boys (20.5 in girls) to the adult obesity cut-off BMI of 30 kg/m^2^ [18]. Based on these cut-offs, overweight was present in 12 children and obesity in six children with frequencies that were not different between groups (Table 4). The percentage of children with a BMI z-score > +1 was 26% in IF, 18% in CF and 19% in BF children (Table 4).

In agreement with the *t*-test and ANCOVA analyses, a linear regression analysis that considered maternal pre-pregnancy BMI, as well as z-scores for length, weight and BMI at the age of one month, showed no significant association of the group allocation, including the breastfed infants, in the main study with the corresponding z-scores at the age of seven years, but 12 to 16% of the variation of the 7-year z-scores were explained by the predictors (Table 5). In further models, insulin or IGF-1 measured at age four months were included as predictors. While insulin did not show any significant associations (data not shown), the IGF-1 values at the age of four months (CF: 55.5 ± 16.4 ng/mL, *n* = 49; IF 62.1 ± 21.2, *n* = 53; BF: 44.6 ± 16.3, *n* = 53) were significantly positively associated with the 7-year z-scores for weight, height and BMI, increasing the explained variation (R^2^-value) to 13–21% (Table 5).

Serious adverse events occurred in 13% (*n* = 7) of children in IF, 16% (*n* = 8) in CF and 9% (*n* = 5) in BF (chi^2^ = 0.522) without a statistically significant difference in the frequency pattern of serious adverse events between groups. Most of the infants were diagnosed with otitis media, pneumonia or problems with tonsils.

## 4. Discussion

Children that were fed a formula with reduced protein and modified protein quality during the first four months did not differ in terms of height, weight, BMI and body composition from those previously fed a standard formula or those who were breastfed. During the period of feeding with IF, with its lower, but improved (addition of α-lactalbumin), protein content and added long-chain polyunsaturated fatty acids, a higher energy efficiency than feeding CF was demonstrated [11]. The 7-year follow-up data were in line with the findings at the age of four years regarding similar growth outcomes. The significant difference in the head circumference gain between CF and BF in the period from four to seven years might have been an incidental finding without biological relevance, as no other status or change comparison of head circumference in the study, including the main study period, showed any significant difference.

Although not confirmed in all studies, meta-analyses showed that anthropometric development and obesity risk is higher for formula-fed than for breastfed children [19,20]. The observed trend toward higher BMI z-score increase from month one to year seven, and especially the period from year four to year seven in formula-fed compared with breastfed children agreed with such a difference.

The follow-up of the BeMIM trial at four and seven years showed no influence of early protein intake and early growth on the childhood obesity risk, which did not confirm the findings in CHOP [8] and observational studies [21]. As not only the amount of protein but also the amino acid pattern were different between the study formulas, our finding is not in contradiction with the CHOP findings but emphasizes that both protein quantity and quality are important for infant growth. While the average growth (changes in the weight-for-age and length-for-age z-scores) during the first four months (intervention period) was higher in IF compared with CF, the growth from four months to seven years was not significantly different between IF and CF, but tended to be lower in IF. Thus, in the follow-up population, a period of higher growth (weight and length) during the first four months was followed by a variable though generally lower growth from four months to seven years in IF infants compared with CF infants, and no difference between both groups was observed at seven years of age. This agreed with the identification of a subgroup of about 40% of children in a combined analysis of several cohorts, which showed non-persistent accelerated early growth, which was not associated with an increased later obesity risk [20].

The anthropometric data collected in the 7-year follow-up compared well with other studies. A study from 2007 in Serbia reported weights and heights that were slightly lower than in our study children, but the BMI (15.7 kg/m^2^ and 16.1 kg/m^2^ for urban boys and girls, respectively) and overweight rate (13.7% and 12.4% for urban boys and girls, respectively) were very similar to our findings [22]. Among 1134 children (age 7–12 years) studied in 2012, 18.9% were overweight, which is very similar to our observed overweight rate of 16% [23]. From 2010 to 2014, the heights and weights of Serbian school children (*n* = 671) were reported to be somewhat lower than in our cohort, with 15.3 kg/m^2^ being the average BMI for boys, similar to our observation, while for girls, they reported 14.7 kg/m^2^, which was slightly lower than our observation of 15.8 ± 2.3 kg/m^2^ [24]. Although the body fat percentage calculated from skinfolds was somewhat higher than that observed in the CHOP study in 6-year-old children (girls 16.4 ± 4.6%, boys 14.8 ± 5.1%), findings agreed that there were no significant differences between study groups, while values were significantly higher for girls than for boys [25].

The follow-up did not indicate any long-term effect of the study group, but regression analyses showed that anthropometry at the age of seven years was significantly related to anthropometry at the age of one month, i.e., shortly after enrolment into the study. This confirmed the importance of adjustment for initial values or birth weight to identify intervention effects in infant feeding studies and agreed with the evaluation procedures in other studies [7,26]. The data agreed with previous findings stating that maternal pre-pregnancy BMI is positively associated with the z-scores for weight and BMI of the children at seven years of age [27]. Of interest is the observation that infants’ diet during the intervention period was not significantly related to anthropometry of the infants at the age of seven years, while the IGF-1 concentrations at the age of four months seemed to partially predict weight, height and, to a lesser extent, BMI. Associations between IGF-1 at the age of nine months and height, weight and BMI at the age of three years were observed in the Danish SKOT cohort [28], and in the Cambridge Baby Growth Study associations between IGF-1 and length gain and BMI change were observed between measures taken at three months and twelve months [29]. Our observations extended the prediction period for height and weight into early school age, but the studies did not agree regarding an effect on BMI. The absence of a detectable effect of four-month insulin levels on later anthropometry could have been due to the high variability in insulin levels with prandial state, which was undefined in our samples. The observation of insulin levels after meals of growing-up milks with different protein content in adults showed that higher protein content is associated with higher insulin, which could have anabolic effects [30]. While the predictive power of metabolite levels for latter anthropometric development has so far been found to be limited in children, it might be that IGF-1 is an important mediator [13,31].

Strengths and Limitations

The strength of the BeMIM follow-up study was the long follow-up duration and the reasonably low attrition rate of the cohort. Measurements were performed according to standardized procedures in the main study and during the follow-up. Nevertheless, it has to be considered a limitation that the statistical power was calculated to detect a difference in growth during the main study until month four of age, but not for the long-term outcomes; therefore, subtle group differences cannot be excluded. Mothers of the 7-year follow-up participants were more highly educated, smoked less and were younger than the mothers of non-participants, but this should only marginally affect the generalizability of the findings. Similarly, although only 54% of the infants completed the initial study according to protocol, this should not affect the interpretation of the follow-up findings.

## 5. Conclusions

The 7-year BeMIM follow-up data suggested that feeding infants with an intervention formula with reduced protein content and added α-lactalbumin enriched whey and long-chain polyunsaturated fatty acids during the first four months of life was safe and did not lead to differences in anthropometry or the occurrence of serious illnesses up to early school age. The importance of such a long-term follow-up of children who were fed infant formulas with different energetic efficiency was emphasized by a recent study that compared a standard infant formula with a formula with improved composition, including the inclusion of α-lactalbumin enriched whey, lactoferrin and whole cow’s milk [32]. In agreement with BeMIM during the intervention period until the age of 24 weeks, higher energetic efficiency for growth was observed for the investigational formula [32], which might suggest that higher energetic efficiency is linked to improved protein quality. Further formulas with lower protein content, but of higher quality, can be expected; therefore, it is reassuring that the 7-year BeMIM follow-up did not find any indications of negative effects of such formulas. Concerning potentially mediating mechanisms, our results led us to conclude that early dietary modulation of IGF-1, but not of insulin, could have a programming effect on long-term growth.

## Figures and Tables

**Figure 1 nutrients-14-03900-f001:**
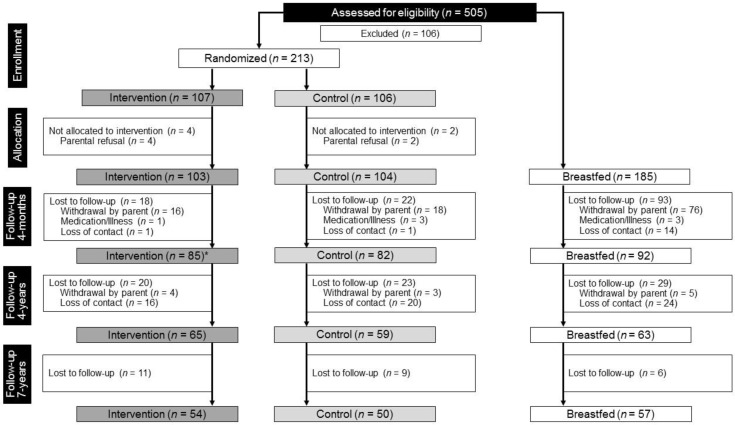
Flow chart according to CONSORT of participants for randomization, allocation and follow-ups at four months, four years and seven years; please note: while, in total, 30 of the 4-year follow-up participants were lost to follow-up, four children not participating in the 4-year follow-up could be included in the 7-year follow-up. * Three children were excluded from the analyses and not invited to the 7-year follow-up.

**Figure 2 nutrients-14-03900-f002:**
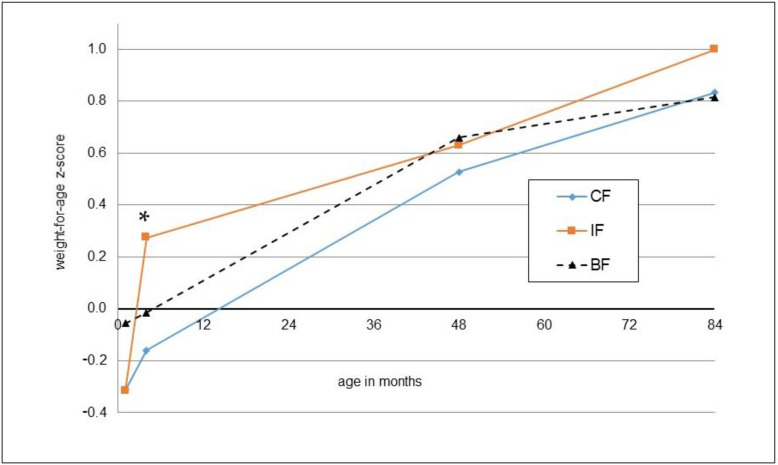
Mean z-scores for weight-for-age at one month (IF: *n* = 50, CF: *n* = 54, BF: *n* = 57), four months (*n* = 50, *n* = 54, *n* = 57), four years (*n* = 48, *n* = 52, *n* = 57) and seven years (*n* = 50, *n* = 54, *n* = 57) for those children participating in the 7-year follow-up; * significant difference between IF and CF, *t*-test.

**Figure 3 nutrients-14-03900-f003:**
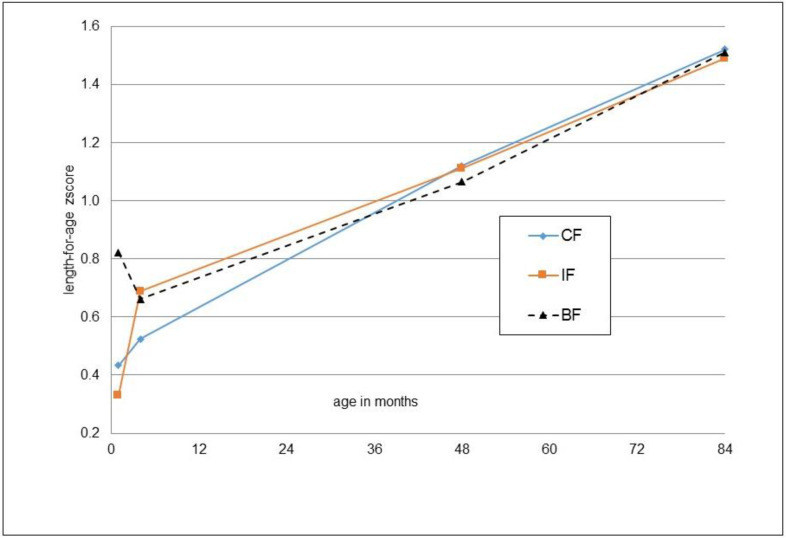
Mean z-scores for length-for-age at one month (IF: *n* = 50, CF: *n* = 54, BF: *n* = 57), four months (*n* = 50, *n* = 54, *n* = 57), four years (*n* = 48, *n* = 52, *n* = 57) and seven years (*n* = 50, *n* = 54, *n* = 57) for those children participating in the 7-year follow-up; no significant difference between IF and CF, *t*-test.

**Figure 4 nutrients-14-03900-f004:**
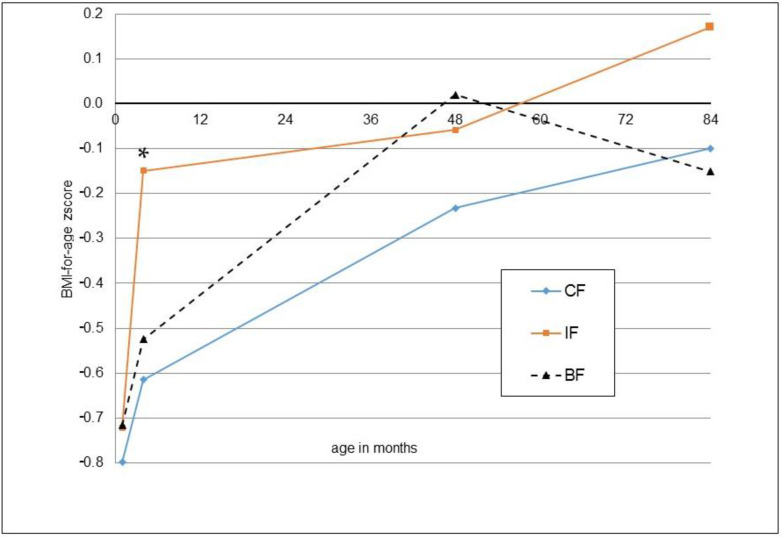
Mean z-scores for BMI-for-age at one month (IF: *n* = 50, CF: *n* = 54, BF: *n* = 57), four months (*n* = 50, 5 *n* = 4, *n* = 57), four years (*n* = 48, *n* = 52, *n* = 57) and seven years (*n* = 50, *n* = 54, *n* = 57) for those children participating in the 7-year follow-up; * significant difference between IF and CF, *t*-test.

**Table 1 nutrients-14-03900-t001:** Description of the infants that participated in the 7-year follow-up (FU) stratified according to study groups: intervention formula (IF), control formula (CF) and breastfed (BF).

	IF	CF	BF
Male sex *n* (%)	28 (51.9)	28 (56.0)	31 (54.4)
Maternal education * Basic/additional/tertiary *n* (%)	1/38/15 (1.9/70.4/27.8)	2/33/15 (4.0/66.0/30.0)	-/27/30 (0/47.4/52.6)
Mother smoked at 7-year FU *n* (%)	20 (37.0)	19 (38.0)	16 (28.1)
Age mother at 7-year FU (years, mean ± SD)	38.5 ± 5.0	39.0 ± 4.9	38.8 ± 4.5
BMI mother at 7-year FU (kg/m^2^, mean ± SD)	23.7 ± 3.5	23.6 ± 4.0	22.9 ± 4.2
Father smoked at 7-year FU *n* (%)	21 (38.9)	16 (32.0)	22 (38.6)
Age father at 7-year FU (years, mean ± SD)	41.3 ± 5.4	41.8 ± 5.9	41.7 ± 6.3
BMI father at 7-year FU (kg/m^2^, mean ± SD)	27.5 ± 5.0	26.8 ± 3.6	28.0 ± 4.8

* Significantly different (chi^2^ test) IF vs. BF and CF vs. BF.

**Table 2 nutrients-14-03900-t002:** Anthropometry and body composition at the age of seven years (mean ± SD) for the interventional formula group (IF), control formula group (CF) and formerly breastfed infants (BF).

	IF (*n* = 54)	CF (*n* = 50)	IF vs. CF ^1^	IF vs. CF ^2^	BF (*n* = 57)	IF vs. BF ^1^	CF vs. BF ^1^
Weight (kg)	27.0 ± 5.2	26.2 ± 4.0	0.377	0.64 [−1.08; 2.36], *p* = 0.465	26.2 ± 5.1	0.382	0.956
Weight-for-age (z-score)	1.00 ± 1.13	0.83 ± 1.01	0.434	0.13 [−0.26; 0.52], *p* = 0.502	0.82 ± 1.16	0.402	0.930
Height (cm)	129.2 ± 5.3	129.5 ± 4.4	0.819	−0.11 [−1.90; 1.69], *p* = 0.907	129.4 ± 5.2	0.901	0.914
Height-for-age (z-score)	1.49 ± 0.99	1.52 ± 0.83	0.860	−0.01 [−0.34; 0.33], *p* = 0.973	1.51 ± 0.96	0.905	0.957
Head circumference (cm)	52.6 ± 1.8	52.8 ± 1.4	0.626	−0.10 [−0.64; 0.44], *p* = 0.705	52.6 ± 1.3	0.855	0.431
BMI (kg/m^2^)	16.1 ± 2.6	15.6 ± 1.7	0.210	0.41 [−0.42; 1.24], *p* = 0.328	15.6 ± 2.5	0.268	0.996
BMI-for-age (z-score)	0.17 ± 1.34	−0.10 ± 1.19	0.280	0.18 [−0.28; 0.64], *p* = 0.433	−0.15 ± 1.44	0.227	0.840
Body fat from BIA (%)	15.5 ± 7.5	14.7 ± 6.3	0.523	0.66 [−2.01; 3.32], *p* = 0.626	13.0 ± 8.3	0.101	0.256
Triceps (mm)	12.1 ± 4.8	11.4 ± 3.8	0.425	0.43 [−1.11; 1.97], *p* = 0.582	11.0 ± 3.7	0.193	0.597
Subscapular (mm)	7.9 ± 4.1	6.8 ± 1.9	0.102	0.90 [−0.31; 2.11], *p* = 0.142	7.3 ± 3.4	0.424	0.391
Body fat from skinfolds (%) ^3^	18.3 ± 6.3	17.2 ± 4.5	0.278	0.86 [−1.13; 2.84], *p* = 0.393	17.2 ± 5.1	0.295	0.992

^1^ *p*-value Student’s *t*-test. ^2^ Estimated difference [95%CI] derived from an ANCOVA adjusted for age (years) of the mother at 7-year follow-up, age (years) of the child at 7-year follow-up, sex, smoking status of the mother at 7-year follow-up and corresponding baseline value, if available. ^3^ Slaughter et al. [16].

**Table 3 nutrients-14-03900-t003:** Growth per day (mean ± SD) during the first seven years of life for the interventional formula group (IF), control formula group (CF) and formerly breastfed infants (BF).

	*n*	IF	*n*	CF	IF vs. CF ^a^	*n*	BF	IF vs. BF ^a^	CF cv. BF ^a^
Weight gain (g/d)									
7 years–1 month	54	9.0 ± 2.0	50	8.7 ± 1.6	0.365	57	8.6 ± 1.9	0.275	0.790
7 years–4 months	54	8.2 ± 2.0	50	8.0 ± 1.6	0.585	57	8.0 ± 2.0	0.495	0.854
7 years–4 years	52	8.1 ± 3.2	48	7.7 ± 2.5	0.572	57	7.4 ± 3.0	0.310	0.593
Height gain (mm/d)									
7 years–1 month	54	0.295 ± 0.020	50	0.295 ± 0.017	1.000	57	0.291 ± 0.019	0.298	0.264
7 years–4 months	54	0.266 ± 0.020	50	0.268 ± 0.017	0.615	57	0.266 ± 0.018	0.928	0.656
7 years–4 years	52	0.192 ± 0.026	48	0.195 ± 0.024	0.485	57	0.196 ± 0.024	0.367	0.856
Head circumference gain (mm/d)									
7 years–1 month	54	0.062 ± 0.006	50	0.062 ± 0.005	0.854	57	0.060 ± 0.007	0.152	0.094
7 years–4 months	54	0.045 ± 0.005	50	0.046 ± 0.005	0.518	57	0.045 ± 0.006	0.881	0.444
7 years–4 years	52	0.014 ± 0.009	48	0.015 ± 0.007	0.327	57	0.011 ± 0.009	0.082	0.004

^a^ *p*-value (Student’s *t*-test).

**Table 4 nutrients-14-03900-t004:** Number (%) of overweight, obese or BMI z-score > +1 children at seven years of age for the intervention group (IF), control group (CF) and formerly breastfed infants (BF).

	IF	CF	IF vs. CF ^a^	BF	IF vs. BF ^a^	CF cv. BF ^a^
Overweight acc. to Cole *n* (%)	12 (22%)	5 (10%)	0.115	9 (16%)	0.470	0.407
Obesity acc. to Cole *n* (%)	3 (6%)	0 (0%)	0.244	3 (5%)	1.000	0.246
BMI z-score > +1	14 (26%)	9 (18%)	0.355	11 (19%)	0.497	1.000

^a^ *p*-value (chi^2^ *t*-test).

**Table 5 nutrients-14-03900-t005:** Linear regression models that describe the association between the anthropometry z-scores at the ages of one month and seven years after considering the effects of diet groups in the main study, maternal pre-pregnancy BMI and IGF-1 levels at four months of age.

Outcome Weight-for-Age z-Score at Seven Years					
Considering maternal pre-pregnancy BMI, one-month weight z-score and diet groups (R^2^, _corr_: 0.16)			Considering maternal pre-pregnancy BMI, one-month weight z-score, diet groups and **IGF-1 at the age of four months** (R^2^, _corr_: 0.21)		
	Beta coefficient ± SE	*p*-value		Beta coefficient ± SE	*p*-value
WZS ^1^ at 1 month	0.473 ± 0.111	3.7 × 10^−5^	WZS ^1^ at 1 month	0.554 ± 0.111	2.0 × 10^−6^
BMI mother	0.072 ± 0.026	0.006	BMI mother	0.069 ± 0.027	0.010
CF *	-	0.140	CF *	-	0.986
IF *	-	0.611	IF *	-	0.756
			IGF-1	0.015 ± 0.004	0.001
Outcome Height-for-Age z-Score at Seven Years					
Considering maternal pre-pregnancy BMI, one-month height z-score and diet groups (R^2^, _corr:_ 0.14)			Considering maternal pre-pregnancy BMI, one-month height z-score, diet groups and **IGF-1 at the age of four months** (R^2^, _corr_: 0.21)		
HZS ^2^ at 1 month	0.372 ± 0.073	8.8 × 10^−7^	HZS ^2^ at 1 month	0.421 ± 0.073	5.2 × 10^−8^
BMI mother	-	0.082	BMI mother	0.049 ± 0.022	0.027
CF *	-	0.366	CF *	-	0.488
IF *	-	0.429	IF *	-	0.733
			IGF-1	0.013 ± 0.004	0.001
Outcome BMI-for-Age z-score at Seven Years					
Considering maternal pre-pregnancy BMI, one-month BMI ^1^ z-score and diet groups (R^2^, _corr_: 0.12)			Considering maternal pre-pregnancy BMI, one-month BMI z-score, diet groups and **IGF-1 at the age of four months** (R^2^, _corr_: 0.13)		
BMIZS ^3^ at 1 month	0.430 ± 0.131	0.001	BMIZS ^3^ at 1 month	0.456 ± 0.131	0.001
BMI mother	0.088 ± 0.032	0.007	BMI mother	0.081 ± 0.034	0.017
CF *		0.871	CF *	-	0.764
IF *		0.207	IF *	-	0.795
			IGF-1	0.012 ± 0.006	0.035

^1^ Weight-for-age z-score, ^2^ height-for-age z-score, ^3^ BMI z-score. * CF—control formula, IF—intervention formula with breastfed (BF) infants as the reference group.

## Data Availability

The authors declared that they will share data for research purposes upon request under conditions respecting the EU General Data Protection Regulation and the protection of the personal rights of study subjects.

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
