# Peer review of "Infant Feeding Choices during the First Post-Natal Months and Anthropometry at Age Seven Years: Follow-Up of a Randomized Clinical Trial"

_nutrients, 2022, doi:10.3390/nu14193900_

Round 1

Reviewer 1 Report

This manuscript reports the findings of the BeMIM study that randomized infants to low protein formula with LPUFA or a control formula. A reference breastfeeding group was also characterized from 1-4 months of age, followed up at age 4 years. The manuscript focuses on follow up of the children at age 7. The manuscript is well written and contributes to the existing literature. Several considerations could improve the manuscript: 

1. Can the authors please add a power analyses and estimate their ability to detect differences?

2. Please add a per protocol analysis to ensures that low compliance doesn't interfere with findings. If results are similar, simply add this in the result section. If not, I suggest reporting both analyses. 

3. Please add head circumference measurement and accuracy details to the method section. It is currently missing

4. Please add details on page 3, line 113 on what data was retrieved from medical records and how safety was assessed. 

5. Please add maternal BMI to the adjusted models/statistics as this has been identified as a confounder for child's overweight/obese status.

6. Please expand the language or add a table of the comparison of completers vs. non completers (page 5, line 163) so that the reader can better understand the differences. 

7. Please statistics to Figure 2, 3, and 4.

8. Please add the adiposity results in tables & results

9. Results have already been published for 1 month to 4 years. Please focus the manuscript only on the new findings. (delete first two raws of table 3). 

10. Page 9 line 219 please add the direction of the association and present results. 

11. Reorganize table 4 to be clearer to reader. 

12. Please define serious illnesses and add details on the results on page 9 line 226.

13. Page 10 line 236 please reword - 'might by a change'

14. Add to the limitations: differences in completers vs. non completers, compliance rate

Author Response

Dear reviewer,

thank you very much for your insightful comments. We really appreciate the time you invested and your thoughts, which help us to improve the manuscript.

Please find below a point by point response to your comments, which we tried to incorporate into the manuscript as far as possible.

  1. Can the authors please add a power analyses and estimate their ability to detect differences?
    The power analysis was based on the main outcome, namely daily growth between birth and 4 months of age. However, the paper describes the results of the 7-year follow-up for which no power analysis was planned/performed and the results should be considered non-confirmatory. The authors have added a sentence in the Statistics section that the results should be considered fully exploratory (lines 145-146). The Strengths and Limitations section contains a sentence on this limitation (lines 311-314).
  2. Please add a per protocol analysis to ensures that low compliance doesn't interfere with findings. If results are similar, simply add this in the result section. If not, I suggest reporting both analyses.
    We agree with the reviewer that per protocol analysis shall be added. Corresponding analyses were performed and yielded very similar findings as the intention to treat population. This is stated now at lines 91-93 and line 175, respectively. For completeness of reporting a table with the per protocol findings has been included into the Supplementary data.
  3. Please add head circumference measurement and accuracy details to the method section. It is currently missing.
    The information about measurement of head circumference, including accuracy, is now given in lines 102-104: “Head circumference was measured with a tape (Seca 212, Hamburg, Germany). All measurements were performed in duplicate and documented with an accuracy of 10 g or 0.1 cm, respectively”.
  4. Please add details on page 3, line 113 on what data was retrieved from medical records and how safety was assessed.
    The collection of safety data is now described in more detail in lines 115-118: “The 7-year follow up included questions and data retrieval from medical records to determine serious illnesses of the children, which could have influenced growth. These illnesses were considered serious adverse events and used to assess long term safety of the study formulas.”
  5. Please add maternal BMI to the adjusted models/statistics as this has been identified as a confounder for child's overweight/obese status
    We agree with the reviewer that maternal BMI is a relevant predictor of infant weight and thank for the idea. We have included maternal pre-pregnancy BMI into the regression models for the prediction of the 7-year anthropometry from infant anthropometry at age 1 month, study group and IGF-1. Indeed we could confirm in our data that maternal BMI is associated with offspring weight and BMI at age seven years (Table 5, lines 290-292).
    As maternal pre pregnancy BMI was not significantly different between the formula groups (IF 22.0±2.9, CF 22.3±3.4, t-test p= 0.653) we did not include this further confounder into the randomized group comparison of anthropometry at age 7 years.
  6. Please expand the language or add a table of the comparison of completers vs. non completers (page 5, line 174) so that the reader can better understand the differences. 
    A detailed comparison of participants and non participants has now been included in the supplementary data, which is referenced in line 170.
  7. Please statistics to Figure 2, 3, and 4.
    We have added to the legends of the figures that means of the z-scores are shown, and statistically significant differences are indicated
  8. Please add the adiposity results in tables & results
    Adiposity results are described in lines 211-214 and for a more clear presentation a table was added to display the results (Table 4).
  9. Results have already been published for 1 month to 4 years. Please focus the manuscript only on the new findings. (delete first two rows of table 3). 
    We fully agree with the reviewer that the findings from the first 4 months are published and there is no need to repeat them. Corresponding lines have been deleted from Table 3.
  10. Page 9 line 219 please add the direction of the association and present results. 
    It has now been added that IGF-1 and z-scores of anthropometry were positively associated (line 227).
  11. Reorganize table 4 to be clearer to reader. 
    The former Table 4 (now Table 5) has been redesigned, avoiding the naming Model 1 and Model 2 and introducing intermediate headlines for the outcomes weight, height and BMI.
  12. Please define serious illnesses and add details on the results on page 9 line 226.
    We thank the reviewer for this comment and take the opportunity to include in lines 115-118 that SAE are meant. The results are given in more detail now in lines 236-239: “Serious adverse events occurred in 13% (n=7) of children in IF, 16% (n=8) in CF and 9% (n=5) in BF (chi2 = 0.522) without a statistical significant difference for frequency pattern of serious adverse events between groups. Most of the infants were diagnosed with otitis media, pneumonia or problems with tonsils.
  13. Page 10 line 236 please reword - 'might by a change
    We have reworded the sentence to “might be an incidental finding without biological relevance” (line 249).
  14. Add to the limitations: differences in completers vs. non completers, compliance rate
    We agree with the reviewer that these points should be mentioned as weak points of the study, but add that we do not assume huge consequences for the interpretation of the findings. Two sentences have been added: “Mothers of the 7-year follow up participants were more highly educated, smoked less and were younger than the mothers of non participants, but this should only marginally affect generalizability of the findings. Similarly, although only 54% of the infants completed the initial study according to protocol, this should not affect the interpretation of the follow up findings.” (lines 314-318).

Reviewer 2 Report

The study is a welcome follow-up on a well-conducted randomized trial. Even though the original study didn't power for the long-term follow-up and only 63% followed up from the original study, the findings are relevant and adds to the literature in the field. The data is presented succinctly with good tables and graphs which are well positioned in the body of the paper.

One suggestion is to include a sentence to say that the bioelectrical impedance analysis is validated for the pediatric population giving the citation

Author Response

Dear reviewer,

thank you very much for your insightful comments. We really appreciate the time you invested and your thoughts, which help us to improve the manuscript.

Please find below a point by point response to your comments, which we tried to incorporate into the manuscript as far as possible.

One suggestion is to include a sentence to say that the bioelectrical impedance analysis is validated for the pediatric population giving the citation
Thank you very much for this suggestion. We have mentioned the validation of BIA and included a corresponding reference [number 17] in lines 112-114.